# Novel Disulfiram-Loaded Metal–Organic Nanoparticles Inhibit Tumor Growth and Induce Immunogenic Cell Death of Triple-Negative Breast Cancer Cells

**DOI:** 10.3390/pharmaceutics17111448

**Published:** 2025-11-09

**Authors:** Chung-Hui Huang, Xuejia Kang, Lang Zhou, Junwei Wang, Shuai Wu, Peizhen Sun, Qi Wang, Adam B. Keeton, Pengyu Chen, Gary A. Piazza

**Affiliations:** 1Department of Drug Discovery and Development, Harrison College of Pharmacy, Auburn University, Auburn, AL 36849, USA; czh0113@auburn.edu (C.-H.H.); jzw0164@auburn.edu (J.W.); qzw0017@auburn.edu (Q.W.); abk0039@auburn.edu (A.B.K.); 2Materials Engineering, Department of Mechanical Engineering, Auburn University, Auburn, AL 36849, USA; xzk0004@auburn.edu (X.K.); lzz0028@auburn.edu (L.Z.); szw0151@auburn.edu (S.W.); pzs0073@auburn.edu (P.S.)

**Keywords:** disulfiram-copper complex [Cu(DDC)_2_], immunogenic cell death (ICD), tumor microenvironment modulation, nanoparticle drug delivery

## Abstract

**Background/Objectives:** Triple-negative breast cancer (TNBC) is among the most aggressive subtypes, lacking estrogen, progesterone, and HER2 receptors, which limits the efficacy of targeted therapies. Standard treatments often fail due to rapid drug resistance and poor long-term outcomes. Repurposing approved drugs with anticancer potential offers a promising alternative. Disulfiram (DSF), an FDA-approved alcohol-aversion drug, forms a copper complex [Cu(DDC)_2_] with potent anticancer activity, but its clinical translation is hindered by poor solubility, limited stability, and inefficient delivery. **Methods:** Here, we present an amphiphilic dendrimer-stabilized [Cu(DDC)_2_] nanoparticle (NP) platform synthesized via the stabilized metal ion ligand complex (SMILE) method. **Results:** The optimized nanocarrier achieved high encapsulation efficiency, enhanced serum stability, and potent cytotoxicity against TNBC cells. It induced immunogenic cell death (ICD) characterized by calreticulin exposure and ATP release, while modulating the tumor microenvironment by downregulating MMP-3, MMP-9, VEGF, and vimentin, and restoring epithelial markers. In a 4T1 TNBC mouse model, systemic [Cu(DDC)_2_] NP treatment significantly inhibited tumor growth without combinational chemo- or radiotherapy. **Conclusions:** This DSF-based metal–organic NP integrates drug repurposing, immune activation, and tumor microenvironment remodeling into a single platform, offering strong translational potential for treating aggressive breast cancers.

## 1. Introduction

Breast cancer remains the most frequently diagnosed malignancy among women in the United States, accounting for approximately 30% of all new female cancer cases annually [1]. Nearly 10–20% of patients lack estrogen, progesterone, or HER2 receptors and are classified as TNBC [2]. TNBC has been identified as a more aggressive subtype with earlier metastasis and overall poorer prognosis compared to other breast cancer subtypes [3]. Standard treatments for TNBC are limited to conventional chemotherapy and surgery. However, chemotherapy is often short-lived in its effectiveness due to rapid development of drug resistance [4]. This therapeutic gap has driven the search for repurposing clinically approved, off-patent drugs with latent anticancer activity.

DSF, originally approved by the FDA as an alcohol-aversion medication, has shown promise in cancer treatment [5]. Upon administration, DSF is rapidly metabolized into diethyldithiocarbamate (DDC), which chelates Cu^2+^ to form a complex [Cu(DDC)_2_]. This complex exhibits cytotoxicity by elevating reactive oxygen species (ROS) and inhibiting proteasome function, offering a potential mechanism to overcome drug resistance in cancer cells [2,3]. However, clinical application of DSF has been limited by poor drug stability, rapid systemic metabolism, and limited plasma half-life [6,7], while direct use of [Cu(DDC)_2_] is impractical due to extremely low solubility (<0.5 ng/mL) [8]. These challenges underscore the need for a robust, stable, and bioavailable delivery system for [Cu(DDC)_2_].

Recent studies have further shown that DSF/Cu co-treatment may induce ICD in breast cancer cells [9], suggesting a promising new avenue for the treatment of TNBC through immunotherapy. ICD is a regulated type of cell death that activates an adaptive immune response in immunocompetent hosts [10]. Cytotoxic chemotherapies such as doxorubicin and oxaliplatin trigger ICD by promoting endoplasmic reticulum (ER) stress and releasing damage-associated molecular patterns (DAMPs) [11]. These DAMPs include CRT, ATP, and high-mobility group box-1 (HMGB1), which act, respectively, as “eat me”, “find me”, and immunostimulatory signals, facilitating dendritic cell maturation and antigen presentation [12,13,14,15]. While DSF/Cu has been shown to promote ICD, it remained unclear whether the pre-formed [Cu(DDC)_2_] complex, particularly when delivered via NP, can elicit a comparable immunogenic response.

Herein, we developed amphiphilic dendrimer-capped [Cu(DDC)_2_] NP to overcome the delivery obstacles and enhance the therapy’s efficacy against TNBC. Encapsulation of [Cu(DDC)_2_] within a dendritic amphiphilic carrier improves its solubility and serum stability, circumvents the rapid metabolism of DSF, and eliminates the need for in vivo copper supplementation, collectively enabling sustained systemic levels of the active complex. To optimize the NP formulation, we synthesized a library of amphiphilic PEGylated lysine-based dendrimers and identified PEG_5k_-Lys_4_-Cholic Acid_8_ (PLCA8) as the optimal stabilizer based on its high drug loading capacity and colloidal stability. Using this optimized PLCA8/[Cu(DDC)_2_] formulation, we demonstrated potent in vitro cytotoxicity against TNBC cells at sub-micromolar IC_50_ concentrations. Importantly, the NP induced hallmark features of ICD, including surface CRT exposure and extracellular ATP release. Beyond their cytotoxic activity, PLCA8/[Cu(DDC)_2_] NP also modulated the tumor microenvironment by downregulating MMP-3 and MMP-9, and key angiogenic and mesenchymal markers such as VEGF, N-cadherin, and vimentin, thereby suppressing angiogenesis and metastatic potential. In vivo studies further demonstrated that PLCA8/[Cu(DDC)_2_] NP effectively suppressed tumor progression in a TNBC mouse model following a single systemic administration, without requiring adjunctive chemotherapy or radiotherapy. These findings demonstrate the potential of this strategy to enhance and broaden anticancer efficacy by integrating targeted cytotoxicity with immune activation, offering a promising avenue for the development of next-generation immunomodulatory nanotherapeutics against TNBC.

## 2. Materials and Methods

### 2.1. Materials

Methoxy PEG Amine HCl salt (PEG5K-NH_2_, Cat. # M-NH2HCl-5000) was purchased from JenKem Technology (Plano, TX, USA). Nα,ε-Bis-Fmoc-L-lysine, Nα,im-Bis-Fmoc-L-histidine, and N,N’-Diisopropylethylamine (DIEA) were obtained from Chem-Impex International, Inc. (Wood Dale, IL, USA). Fmoc-D-His(Boc)-OH was purchased from BACHEM, and cholic acid was obtained from Beantown Chemical. 1-Hydroxybenzotriazole hydrate (HOBt) was purchased from Sigma-Aldrich (St. Louis, MO, USA), and N,N’-Diisopropylcarbodiimide (DIC) was purchased from Alfa Aesar (MA). All other general chemicals and reagents were obtained from VWR. ELISA kits for IL-12 and IL-1β were purchased from R&D Systems (Minneapolis, MN, USA), and the HMGB1 ELISA kit was obtained from Arigo Biolaboratories (Zhubei City, Taiwan). The following cell lines were used in this study: 4T1 mouse mammary carcinoma cell line (ATCC), U937-DC-SIGN (CRL-2539, ATCC), primary CD8^+^ cytotoxic T cells (PCS-800-017™, ATCC), THP-1 Dual™ NF-κB/IRF reporter cells (CRL-3253™, ATCC), and DC2.4 mouse dendritic cell line (immortalized; Cat. # SCC142, Sigma-Aldrich (St. Louis, MO, USA)).

### 2.2. PEG-Lysine Dendrimer

The PEG_5K_-Lysine dendrimer was synthesized via stepwise peptide coupling process starting from MeO-PEG-NH_2_·HCl [16]. PEG_5K_-NH_2_ (1 g, 0.2 mmol), Fmoc-Lys(Fmoc)-OH (236.4 mg, 0.4 mmol), DIC (75 μL, 0.48 mmol), HOBt (64.8 mg, 0.48 mmol), and DIEA (175 μL, 1 mmol) were dissolved in DMF (25 mL) and stirred at room temperature for 3 h. Completion of the coupling reaction was confirmed by the Kaiser test: yellow indicated the completed reaction with no free amino groups, while a blue color indicated the presence of unreacted amino groups. The reaction mixture was poured into cold diethyl ether (100 mL), and the precipitate was collected by centrifugation, washed with cold ether (3 × 75 mL), and dried under vacuum. Fmoc groups were removed by treatment with 20% piperidine in DMF (*v*/*v* 1:4, 50 mL), and the deprotected products were precipitated and washed with cold ether. This coupling and deprotection process was repeated twice with Fmoc-Lys(Fmoc)-OH to generate the final PEG-lysine-dendrimer structure. After the final deprotection, NHS-Cholic acid was conjugated to the terminal amino groups of the lysine residues to yield PLCA8. The product was further purified by dialysis with deionized water for 2 days and lyophilized to obtain the white powder. The chemical structure and degree of substitution of PLCA8 were confirmed by ^1^H NMR (DMSO-d_6_) (Appendix A).

### 2.3. Preparation of NP

The [Cu(DDC)_2_] NP was prepared with SMILE technology as described in the previous report [17]. In brief, the stabilizer was dissolved in dichloromethane. The dichloromethane was removed by a rotavapor, and the resulting film was dispersed in deionized water. Then, DDC-Na and CuCl_2_ were dissolved in the aqueous solution containing 1% (*w*/*v*) stabilizer to form a DDC-Na solution and a CuCl_2_ solution. [Cu(DDC)_2_] NP was formed by mixing DDC-Na solution and CuCl_2_ solution and vortexing for 1 min. To remove large aggregation, the forming [Cu(DDC)_2_] NP was centrifuged at 10,000 rpm (6700× *g*) for 10 min and filtered through a 0.45 µm membrane.

### 2.4. Characterization of NP

DLS (Zetasizer Nano ZS, Malvern, UK) was used to determine the particle size of [Cu(DDC)_2_] NP. The morphology and elemental compositions of NP were characterized by FESEM using JBM-7000F (JEOL, Tokyo, Japan), equipped with an EDX detector. The concentration of [Cu(DDC)_2_] NP was estimated by a UV spectrometer (Nanodrop 2000, Thermo Scientific, Waltham, MA, USA) at 435 nm. The [Cu(DDC)_2_] concentration was calculated by a standard curve generated with various [Cu(DDC)_2_] concentrations. To determine the stability of [Cu(DDC)_2_] NP. The NP was incubated with 10% fetal bovine serum (FBS) at 37 °C, and the particle size and concentration were examined at different time points. The loading efficiency of [Cu(DDC)_2_] NP was calculated by the following equation: Loading efficiency (%) = (real drug concentration/theoretical drug concentration) × 100%.

The encapsulated [Cu(DDC)_2_] concentration was quantified by UV–Vis spectroscopy at 435 nm after complete nanoparticle disruption in DMSO. As shown in Appendix A, the PLCA8 polymer matrix displayed negligible absorbance at this wavelength, indicating minimal background interference. The calibration curve for {Cu(DDC)_2_] in DMSO followed the linear regression equation y = 0.02989x + 0.003463 with R^2^ = 0.9994 (Appendix A). Free drug was removed by centrifugation prior to measurement, and the remaining pellet was redispersed in DMSO for analysis. Results are reported as mean ± SD (*n* = 3 technical replicates).

### 2.5. Cell Viability Assay

4T1 murine breast cancer cells were cultured in a medium composed of RPMI-1640 (Corning) with 10% FBS and 1% antibiotic-antimycotic. Cells were cultured at 37 °C in a humidified atmosphere containing 5% CO_2_. Cells were seeded into a 96-well plate at a density of 5000 cells/well and incubated overnight. Followed by treating the cells with a series of different concentrations of [Cu(DDC)_2_] NP diluted in a cell culture medium for 48 h, cytotoxicity was determined with the MTT assay. The absorbance was determined with a microplate spectrophotometer. The IC_50_ was calculated with GraphPad software based on a dose–response curve. Due to the poor aqueous solubility of [Cu(DDC)_2_], the free compound was first dissolved in DMSO to prepare a stock solution and diluted in cell culture medium immediately before use (final DMSO ≤ 0.1%). The nanoparticle suspensions were diluted in the same medium to match the molar [Cu(DDC)_2_] concentrations used for free-drug treatments, ensuring equivalent [Cu(DDC)_2_] exposure across groups. 

### 2.6. Calcein-AM and PI Staining

Cells were seeded at a density of 5000 cells per well in a 96-well plate overnight. This was followed by treating cells with different compounds for 24 h and then staining them with a solution composed of Calcein-AM and PI in pH 7.4 PBS. Cell samples were analyzed with a Cytation 5 Cell Imaging Multi-Mode Reader. Viable and dead cells can be identified by the green fluorescence (live) and the red fluorescence (dead), respectively.

### 2.7. HMGB1 Detection

4T1 cells were seeded at a density of 1 × 10^5^ cells per well in a 12-well plate. After incubation overnight, the cells were treated with different test agents for 18 h. Then, the cells were washed with cold PBS twice, fixed with 4% paraformaldehyde for 20 min, and permeabilized with 0.1% Triton X-100 for 10 min. Next, cells were incubated with primary HMGB1 antibody (3935S, Cell signaling) for 1 h and followed by incubating with FITC-conjugated secondary antibody for 1 h. Finally, the cells were stained with DAPI and examined by microscope (NIKON, Tokyo, Japan). The releasing level of HMGB1 were quantified from culture supernatants collected from wells seeded at equal initial cell densities and treated under identical conditions. Samples were not normalized to cell number because equal plating and comparable viability across groups ensured equivalent total cell input. The concentrations of HMGB1 in supernatants were measured using HMGB1 ELISA Kit according to the manufacturer’s instructions.

### 2.8. Calreticulin (CRT) Detection

4T1 breast cancer cells were seeded at a density of 1 × 10^5^ cells per well in a 12-well plate. After being incubated overnight, the cells were treated with different agents for 12 h. Then, cells were harvested by 0.25% trypsin, washed with PBS (0.5% BSA) three times. Cells were incubated with anti-CRT (ab2907, Abcam, Cambridge, UK) primary antibodies for 1 h at RT and the FITC labeled secondary antibodies with for 1 h at RT. The FITC (green) fluorescence intensity in each sample was detected and quantified by flow cytometry (BD Accuri C6, San Jose, CA, USA).

### 2.9. ROS Evaluation

Intracellular ROS levels were measured using the DCFH-DA fluorescence assay. Briefly, cells were seeded in a 24-well plate at a density of 50,000 cells per well. After treatment with the indicated formulations, cells were incubated with 5 µM DCFH-DA in serum-free medium at 37 °C for 30–45 min. Following incubation, cells were washed with PBS to remove excess dye. The fluorescence intensity was measured using a Cytation 5 Cell Imaging Multi-Mode Reader (Agilent Technologies, Santa Clara, CA, USA) with excitation at 485 nm and emission at 535 nm.

### 2.10. ATP Detection

The level of extracellular ATP was determined with an ATP determination assay (A22066, Thermo Scientific, Waltham, MA, USA) according to the manufacturer’s instructions. In brief, 4T1 breast cancer cells were seeded at a density of 1 × 10^4^ cells per well in a 96-well plate. After being incubated overnight, the cells were treated with different agents for 12 h. Then, the conditioned medium was collected and examined with the ATP determination kit. The luminescence was measured using GloMax 96 Microplate Luminometer (Promega Corporation, Madison, WI, USA).

### 2.11. Anti-Angiogenesis Evaluation

To evaluate the effect of NP on angiogenesis, the mouse endothelial cell line BEND3 was selected as the detection model. The control group consisted of a co-culture of 5000 cancer cells, 5000 BEND3 cells, and 5000 macrophages. For the [Cu(DDC)_2_] treatment group, cells were incubated with [Cu(DDC)_2_] for 6 h. After treatment, the NP was removed, and the cells were further incubated for an additional 18 h. Following incubation, culture supernatants were collected and analyzed using ELISA to quantify levels of mouse VEGF-A, MMP-3, MMP-9, sICAM-1, and ANGPT1. Commercial ELISA kits (Abcam, Cambridge, UK) were used according to the manufacturers’ protocols. Specifically, mouse VEGF-A was measured using the Mouse VEGF-A ELISA Kit (Abcam, Cambridge, UK, Cat# ab100751), MMP-3 using the Mouse MMP-3 ELISA Kit (Abcam, Cambridge, UK, Cat# ab100731), MMP-9 using the Mouse MMP-9 ELISA Kit (Abcam, Cambridge, UK, Cat# ab100610), and sICAM-1 using the Mouse ICAM-1 ELISA Kit (Abcam, Cambridge, UK, Cat# ab100688). Mouse ANGPT1 levels were quantified using the Mouse ANGPT1/Angiopoietin-1 Sandwich ELISA Kit (LSBio, Seattle, WA, USA, Cat# LS-F31477). For each assay, a standard curve was generated by serial dilution of the corresponding recombinant mouse protein. Sample concentrations were calculated by interpolating the absorbance values measured at 450 nm against the respective standard curves.

Immunofluorescence staining for MMP-3 was performed using a mouse monoclonal anti-MMP-3 antibody (purchased from Abcam, Cambridge, UK, Cat# ab53015), followed by incubation with an appropriate fluorophore-conjugated anti-mouse secondary antibody. Cells or tissue sections were fixed, permeabilized, and blocked prior to antibody incubation, following standard immunofluorescence protocols.

### 2.12. Western Blot

Mouse 4T1 cancer cells were seeded in a 6-well plate at a cell density of 4 × 10^5^ cells/well. After incubating the cells overnight, the cells were treated with different agents. After 6 h of incubation, cell lysates were prepared using lysis buffer. The cell lysates were separated using sodium dodecyl sulfate-polyacrylamide gel electrophoresis and transferred to membranes for Western blot analysis. Membranes were probed with one of the following primary antibodies: anti-HSP70 (Enzo, Farmingdale, NY, USA, 1:4000), anti-ubiquitin (Cell Signaling, Danvers, MA, USA, 1:1000), anti-CHOP (Proteintech, Rosemont, IL, USA, 1:1000), anti-b-actin (Cell Signaling, Danvers, MA, USA, 1:2000) followed by detection by horseradish peroxidase-conjugated secondary antibodies: goat anti-rabbit IgG-HRP (Cell Signaling, Danvers, MA, USA), anti-mouse IgG-HRP (Santa Cruz, Dallas, TX, USA). After removing secondary antibodies, the membranes were washed with TBS-T (TBS with 0.1% Tween 20). Then, membranes were visualized by an ECL (32209, Thermo Scientific, Waltham, MA, USA) detection reagent and imaged using a Western Blot imaging system (c600, Azure, Dublin, CA, USA).

### 2.13. In Vivo Tumor Growth Inhibition Study

Balb/c mice (female, 7 weeks) were purchased from the Jackson Laboratory (Bar Harbor, ME, USA). The orthotopic breast tumor animal models were established by injecting 4T1 cells (2 × 10^6^ cells) into the first mammary fat pad of the Balb/c mice. Treatments were initiated when the tumors reached an average volume of 100 mm^3^. Tumor size (mm^3^) = [width (mm)^2^× length (mm)] × 1/2. Mice were randomly divided into two groups (6 mice per group) and were given an intravenous injection of [Cu(DDC)_2_] NP and 10% (*w*/*v*) PLCA8 polymer in PBS (control). [Cu(DDC)_2_] NP was administered intravenously at 3 mg/kg, calculated based on the total Cu(DDC)_2_ complex, on days 7, 9, 11, 13, and 15 post-tumor inoculation. The group size and dosage were selected to ensure adequate statistical power and to account for potential animal loss during the study, consistent with prior Cu-based nanoparticle studies employing comparable sample sizes [18]. Body weight and tumor volume were measured and recorded. The mice were euthanized, and the weight of the collected tumors was determined at the end of the experiment. All animal procedures were approved and conducted in accordance with the guidelines of Auburn University’s Institutional Animal Care and Use Committee (protocol #2022-3824).

### 2.14. Immune Cells Co-Culture Experiment

The 4T1 breast cancer cells were seeded in 12-well plates at a cell density of 5 × 10^4^ cells/well and incubated overnight. Cells were treated with different agents for 12 h, and BMDCs were added after the removing of treatment agents in each well. After co-culture for 48 h, the maturation of DCs was examined by determining the concentration of IL-12 and IL-1b in the conditioned cell culture medium with ELISA. Additionally, the levels of CD80, CD86, and CD11c were determined with standard IF protocol. The activated T cells were evaluated with the secretion of Granzyme B, Perforin, and IFN-γ. Those molecules were determined by ELISA kits.

### 2.15. Caspase-3 Activity

Caspase-3 was quantified using a colorimetric DEVD-pNA assay (Abs 405 nm). Cells were seeded at 3 × 10^4^ per well in 96-well plates and treated with [Cu(DDC)_2_] NP; staurosporine (1 µM, 4 h) served as the positive control. After treatment, cells were lysed on ice with lysis buffer for 15 min and clarified by brief centrifugation. Equal volumes of lysate and DEVD substrate were mixed and incubated for 60 min at 37 °C, and signals were read as absorbance at 405 nm for DEVD-pNA.

### 2.16. Statistical Analysis

Data were analyzed by GraphPad Prism 10.4.1 using the unpaired t-test or ANOVA analysis.

## 3. Results

### 3.1. Preparation and Characterization of [Cu(DDC)_2_] NPs

Copper-based therapies have drawn increasing attention in cancer treatment due to their potent cytotoxic effect and their ability to induce oxidative and proteotoxic stress in cancer cells. A key challenge in formulating effective copper-based NP lies in the selection of a suitable stabilizer that enables high drug loading and maintains particle stability in physiological conditions. To this end, we explored PEG-lysine-based dendrimers with varied hydrophobic moieties as stabilizing platforms, leveraging their favorable physicochemical properties and prior success in delivering chemotherapeutic agents [16,17].

Among the dendrimer variants evaluated, the PLCA8 polymer, an amphiphilic PEGylated lysine dendrimer with multiple cholic acid groups, demonstrated superior self-assembly behavior and enhanced NP stability (Figure 1A). The chemical structure and cholic-acid substitution of PLCA8 were verified by ^1^H NMR (Appendix A), confirming successful dendrimer synthesis prior to NP formation. At a molar ratio of 2:1 of DDC^−^ to Cu^2+^, the components formed a water-insoluble complex (Figure 1B). PEG-lysine dendrimer could effectively stabilize in situ [Cu(DDC)_2_] NP and prevent the formation of large aggregations. PLCA8 allowed [Cu(DDC)_2_] encapsulation at a concentration up to 2 mg/mL, achieving near-complete loading efficiency (~100%). In contrast, other stabilizers, including PEG_5k_-lysine_4_-fmoc_8_ (PLF8), PEG_5k_-lysine_4_-(histidine-fmoc)_8_ (PL(H1)8), and PEG_5k_-lysine_4_-(histidine-fmoc_2_)_8_ (PL(H2)8) exhibited significant lower encapsulation efficiency (Figure 1C). Notably, drug loading was not significantly affected by variations in the number of cholic acid groups (Figure 1D). However, stronger hydrophobic interactions resulted in more stable NP formulations. For example, PEG_5k_-Lys-Cholic Acid_2_-based (PLCA2) NP exhibited rapid destabilization, showing significant drug release and size variation after just 2 h of incubation in a serum-containing buffer at 37 °C (Appendix A), while PEG_5k_-Lys_2_-Cholic Acid_4_ (PLCA4) and PLCA8 maintained their structural integrity and retained their payloads for up to 24 h. Comparative studies revealed that PLCA8 outperformed other stabilizers (including PLF8, PL(H1)8, and PL(H2)8) in terms of encapsulation efficiency and serum stability (Appendix A). While all formulations yielded NP with sizes ranging from 40–80 nm as measured by dynamic light scattering (DLS), those stabilized by PLF8 displayed larger particle sizes (~80 nm) and poor drug retention, and were therefore excluded from subsequent experiments (Figure 1E). The Elemental analysis via energy-dispersive X-ray spectroscopy (EDX) confirmed the presence of Cu and S within the NP (Figure 1F,H). Morphological characterization using field emission scanning electron microscopy (FESEM) further validated their nanoscale uniformity and spherical shape (Figure 1G). PLCA8-stabilized NP consistently showed a diameter of ~40 nm with minimal size variation over time and better long-term stability, supporting their suitability for enhanced permeability and retention (EPR)-based tumor targeting (Figure 1I). The hydrodynamic size of PLCA8-stabilized [Cu(DDC)2] NP was further characterized by DLS analysis (Appendix A). The nanoparticles exhibited a narrow monomodal size distribution with an average diameter of 46 ± 0.35 nm, a low PDI of 0.116 ± 0.012, and a zeta potential of −3.12 ± 1.24 mV, confirming uniformity and colloidal stability under aqueous conditions. Together, these results identify PLCA8 as an optimal stabilizer for [Cu(DDC)_2_] NP formulation, achieving high drug loading, serum stability, and well-defined nanoscale morphology, thereby enabling downstream biological and therapeutic evaluations.

### 3.2. [Cu(DDC)_2_] NP Induced ROS-Driven ER Stress via Proteostasis Disruption

To investigate whether [Cu(DDC)_2_] NPs reproduce the stress responses as observed in previously reported DSF/Cu complex, we examined their ability to disrupt proteostasis and trigger ER stress in TNBC cells. [Cu(DDC)_2_] is known to bind the segregase adaptor NPL4, leading to the formation of NPL4-p97 aggregates and disrupting the p97-NPL4-UFD1 complex, a critical regulator of the ER-associated degradation (ERAD) pathway (Figure 2A,B). This disruption hampers the clearance of misfolded or polyubiquitinated proteins, resulting in ER stress and potential apoptosis.

Western blot analysis confirmed that treatment with [Cu(DDC)_2_] NP led to marked accumulation of polyubiquitinated proteins, indicative of proteasome dysfunction (Figure 2B). In parallel, we observed upregulation of heat shock protein 70 (HSP70) and C/EBP homologous protein (CHOP) (key markers of protein misfolding stress and unresolved ER stress, respectively), supporting that [Cu(DDC)_2_] NP disrupted proteostasis and initiated ER stress-mediated apoptosis. To evaluate oxidative stress, we measured intracellular ROS levels using a fluorescence-based assay. Cells treated with [Cu(DDC)_2_] NP showed a significant increase in ROS levels compared to untreated controls and blank NP (Figure 2C and Appendix A). To probe the downstream oxidative stress damage, we quantified malondialdehyde (MDA), a byproduct of lipid peroxidation (Figure 2D). [Cu(DDC)_2_] NP treatment led to significantly increased MDA levels [19], while co-treatment with the antioxidant N-acetylcysteine (NAC) reduced MDA accumulation (Figure 2D), confirming ROS-mediated membrane damage. To evaluate the therapeutic potential of [Cu(DDC)_2_] NPs, we first assessed their cytotoxicity in vitro using the 4T1 murine breast cancer cells. 3-(4,5-dimethylthiazol-2-yl)-2,5-diphenyl tetrazolium bromide (MTT) assay revealed dose-dependent cytotoxicity with half-maximal inhibitory concentration (IC_50_) of approximately 230 nM after 48 h of treatment, while control treatments with free copper, DDC alone, or blank NP showed minimal toxicity (Figure 2E). Beyond population-level viability assays, we visualized individual cell responses using Calcein-AM/propidium iodide (PI) dual staining, which distinguishes live (Calcein-AM-positive, green) from dead cells (PI-positive, red). As shown in Figure 2F, untreated control cells exhibited strong green fluorescence with minimal PI signal, indicating high viability. Similarly, cells treated with sodium diethyldithiocarbamate (DDC-Na), CuCl_2_, or blank NP resulted in negligible cytotoxicity. In contrast, [Cu(DDC)_2_] NP-treated cells displayed intense red PI fluorescence and markedly reduced Calcein-AM signal, confirming significant loss of membrane integrity and extensive cell death. The [Cu(DDC)_2_] efficacy is comparable to that of chemotherapies such as PTX and DOX, as evidenced by the dramatic decrease in cell viability (Appendix A). The anti-cancer effects of [Cu(DDC)_2_] NP are more related to above mechanism rather than caspase3-mediated apoptosis in this study (Appendix A). These results provide both visual and quantitative evidence of the selective and potent tumoricidal activity of [Cu(DDC)_2_] NP through a multifaceted mechanism, involving proteasome inhibition, oxidative stress, lipid peroxidation, and sustained ER stress.

### 3.3. [Cu(DDC)_2_] NP Induced ICD and Enhance T Cell-Mediated Anti-Tumor Immunity

While [Cu(DDC)_2_] NP exhibited potent cytotoxic effects, durable anti-tumor responses in breast cancer require engagement of the host immune system. To evaluate whether [Cu(DDC)_2_] NP can elicit ICD, a form of regulated cell death that stimulates adaptive immunity, we assessed three hallmark indicators of ICD using 4T1 breast cancer cells as the model system: surface exposure of CRT, release of HMGB1, and extracellular ATP secretion.

CRT is an endoplasmic reticulum-resident chaperone protein that, upon ICD induction, translocates to the cell surface and acts as an “eat me” signal to promote the uptake of dying cancer cells by antigen-presenting cells. Following 12 h of [Cu(DDC)_2_] NP treatment, we observed a marked increase in the mean fluorescence intensity of CRT on the 4T1 cell membrane (Figure 3A,B), indicating active CRT translocation and effective induction of ICD. Another key marker, HMGB1, is a nuclear protein that functions as a DAMP when released into the extracellular space. We found that [Cu(DDC)_2_] NP treatment significantly increased the extracellular levels of HMGB1 (Appendix A), with a corresponding decrease in intracellular concentration (Appendix A). This shift supports the release of HMGB1 from dying cells, a process known to facilitate antigen presentation and immune activation. Furthermore, [Cu(DDC)_2_] NP treatment increased extracellular ATP release (Figure 3C). Since extracellular ATP serves as a “find me” signal that recruits and activates dendritic cells (DCs), this observation further confirms the occurrence of ICD.

To determine whether this ICD signature could activate antigen-presenting cells, we performed a DC maturation assay using bone marrow-derived DCs co-cultured with [Cu(DDC)_2_]-treated 4T1 tumor cells (Figure 3D). Immunofluorescence staining was performed to assess maturation markers, where CD11c (green) identifies DCs, CD80 (red) indicates maturation, and 4’,6-diamidino-2-phenylindole (DAPI, blue) stains nuclei. Co-incubation with [Cu(DDC)_2_]-treated 4T1 tumor cells significantly promoted DC maturation, as indicated by increased expression of co-stimulatory surface markers CD80 and CD86 (Figure 3E and Appendix A). Following immunogenic cell death, dendritic cell (DC) maturation was evident. As shown in Figure 3F, DCs exhibited increased MHC class II (MHC II) expression after treatment, indicating enhanced antigen-presenting capacity. Consistently, pro-inflammatory cytokine profiling revealed a concomitant rise in IL-12 levels (Figure 3G), a hallmark of mature, Th1-polarizing DCs that promotes IFN-γ–driven cytotoxic T-cell priming. Together, the upregulated MHC II and elevated IL-12 support a treatment-induced shift toward effective cross-priming and anti-tumor immunity. Additionally, the level of interferon-gamma (IFN-γ), a hallmark cytokine produced by activated T cells, was significantly upregulated in the [Cu(DDC)_2_]-treated group (Figure 3H), indicating robust T cell activation. Importantly, in the presence of NAC, a ROS scavenger, IFN-γ production was markedly reduced (Appendix A), suggesting that ROS generated by [Cu(DDC)_2_] NP plays a critical role in promoting T cell-mediated immune responses. Furthermore, [Cu(DDC)_2_] treatment led to a dose-dependent release of interleukin-12 (IL-12), a cytokine produced by mature DCs that supports T helper 1 cells polarization and cytotoxic T cell activation (Appendix A).

We then evaluated expression of cytotoxic effector molecules to assess T cell function. [Cu(DDC)_2_] NP treatment with significantly upregulated the Granzyme B expression (Figure 3I), a key marker of T cell activation and immune-mediated tumor cell killing, suggesting enhanced cytotoxic T lymphocyte (CTL) activity within the tumor microenvironment. Perforin expression was also elevated (Figure 3I), which acts synergistically with Granzyme B by forming pores in target cell membrane, thereby facilitating Granzyme B entry and apoptosis induction. Together, the upregulation of Granzyme B and Perforin confirms a robust cytotoxic immune response and supports the role of [Cu(DDC)_2_] NP in enhancing T cell-mediated anti-tumor immunity. In parallel, similar upregulation of Granzyme B and Perforin was observed in co-culture experiments with EMT6 tumor cells (Appendix A), further validating the immune-stimulatory potential of [Cu(DDC)_2_] NP.

Co-treatment with brefeldin A (B7651, Sigma-Aldrich, St. Louis, MO, USA) reduced secreted IFN-γ, the possible reason is brefeldin A diminished [Cu(DDC)_2_] NP-induced calreticulin surface exposure, thereby attenuating the downstream ICD cascade (Figure 3J). These results support the immune cell activation of [Cu(DDC)_2_] NP is associated with ICD effects. Collectively, these findings confirm that [Cu(DDC)_2_] NP promoted both DCs maturation and T cell effector function, contributing to a coordinated anti-tumor immune response.

### 3.4. [Cu(DDC)_2_] NP Suppressed Epithelial-Mesenchymal Transition (EMT) and Angiogenesis to Inhibit Tumor Progression

Angiogenesis and EMT are two interconnected processes that play pivotal roles in tumor progression and metastasis. Our [Cu(DDC)_2_] can decrease cell migration; therefore, we further investigated the underlying anti-metastasis mechanism (Appendix A). Angiogenic factors degrade the extracellular matrix (ECM) to facilitate new blood vessel formation, which in turn supports tumor growth and can promote EMT. During EMT, epithelial tumor cells lose their apical–basal polarity and cell–cell adhesion, acquiring mesenchymal traits such as enhanced motility and invasiveness. As illustrated in Figure 4A, this transition is critical in the metastatic spread of cancer cells.

To investigate whether [Cu(DDC)_2_] NP can interfere with these processes, we examined their effects on angiogenic and EMT-related markers in tumor-stromal co-culture models. [Cu(DDC)_2_] NP treatment led to a strong suppression of key pro-angiogenic mediators. VEGF, a key driver of neovascularization, was markedly reduced in the co-culture system of cancer cells and endothelial cells (Figure 4B). Similarly, platelet-derived growth factor (PDGF), which promotes vascular remodeling and stabilization, was downregulated following treatment (Figure 4C). Immunofluorescence analysis also revealed decreased expression of von Willebrand factor (vWF), a glycoprotein involved in angiogenesis and vascular integrity (Figure 4D). Soluble intercellular adhesion molecule-1 (sICAM-1), which is associated with inflammation and ECM remodeling, was also attenuated by [Cu(DDC)_2_] treatment (Figure 4E and Appendix A), suggesting disruption of pro-metastatic signaling. Additionally, [Cu(DDC)_2_] significantly reduced the expression of angiopoietin-1, another pro-angiogenic factor implicated in tumor vascularization and metastasis (Figure 4G). At the molecular level, [Cu(DDC)_2_] NP also modulated several EMT markers. Specifically, the treatment led to the upregulation of E-cadherin, an epithelial adhesion molecule that suppresses invasion and metastasis (Figure 4H). In contrast, N-cadherin (Figure 4I) and Vimentin (Figure 4J), mesenchymal markers typically linked to increased motility and invasiveness, were downregulated. This pattern of gene expression indicates the inhibition of EMT and a potential shift toward a mesenchymal-to-epithelial transition (MET), thereby reducing cellular motility and invasiveness. These findings highlight the dual action of [Cu(DDC)_2_] NP in targeting both angiogenesis and EMT, two tightly linked processes central to tumor metastasis. By simultaneously impairing tumor vascular support and invasive capacity, the NP exerted broad anti-tumor effects beyond direct cytotoxicity.

Additionally, in the tumor microenvironment, transforming growth factor-beta (TGF-β) is known to promote metastasis by upregulating MMPs. In our study, [Cu(DDC)_2_] NP effectively suppressed TGF-β-induced MMP-3 expression (Appendix A), further suggesting that the NP can counteract pro-metastatic signaling pathways. Collectively, these results support the role of [Cu(DDC)_2_] NP in remodeling the tumor microenvironment and limiting metastatic potential through coordinated suppression of EMT and angiogenic signaling.

### 3.5. [Cu(DDC)_2_] NPs Exhibited Potent in Vivo Anti-Tumor Efficacy and Immunomodulatory Effects in a TNBC Mouse Model

To evaluate the in vivo therapeutic potential of [Cu(DDC)_2_] NPs, we assessed their anti-tumor efficacy using the 4T1 murine model of TNBC. Once tumor volumes reached approximately 100 mm^3^, mice were randomly assigned to receive either phosphate-buffered saline (PBS with 10% PLCA8, control) or [Cu(DDC)_2_] NP at a dosage of 3 mg/kg on days 7, 9, 11, 13, and 15 (Figure 5A). In the control group, tumors continued to grow rapidly. In contrast, mice treated with [Cu(DDC)_2_] NP showed a significant reduction in tumor growth (Figure 5B). Importantly, no significant changes in body weight were observed in the treatment group, suggesting favorable biosafety and tolerability of the NP formulation in vivo (Figure 5C). At the end of the treatment period, excised tumors from [Cu(DDC)_2_]-treated mice were visibly smaller than those in the control group (Figure 5D). Tumor weights were measured post-euthanasia, showing a 53.2% decrease in the [Cu(DDC)_2_] NP group compared to the control (Figure 5E). This translated to a 61% reduction in tumor volume relative to controls, demonstrating the substantial therapeutic benefit by [Cu(DDC)_2_] NP.

To assess whether the in vivo anti-tumor effect was associated with immune activation, we examined CRT expression in tumor tissues. Immunofluorescence staining revealed a pronounced increase in CRT levels in tumors from [Cu(DDC)_2_] NP-treated mice (Figure 5F), consistent with the induction of ICD observed in vitro. This increase in CRT suggests enhanced tumor immunogenicity and supports the immunomodulatory role of [Cu(DDC)_2_] NP in vivo. These results demonstrate that [Cu(DDC)_2_] NP not only exerted potent anti-tumor effects in a TNBC mouse model but also enhanced immunogenic features within the tumor microenvironment. Their ability to combine direct tumor suppression with immune activation highlights the translational potential of this NP system for the treatment of aggressive breast cancers. NPL4 levels in tumor tissues were elevated following [Cu(DDC)_2_] NP treatment (Appendix A). For metastasis, immunofluorescence staining showed that [Cu(DDC)_2_] treatment markedly decreased N-cadherin signal intensity and continuity relative to controls, while nuclear DAPI signal remained largely unchanged; accordingly, the merged images displayed reduced magenta–blue colocalization and a focal region of low cell density (Appendix A).

## 4. Discussion

The rational engineering of metal–organic nanotherapeutics offers an opportunity to overcome long-standing challenges in drug delivery and cancer treatment. Selecting an appropriate stabilizer is essential for achieving high encapsulation efficiency and NP stability [20]. By applying the SMILE approach, we successfully formulated [Cu(DDC)_2_] NP using an amphiphilic PLCA8 with cholic acid moieties. PLCA8 provided critical nanoscale stabilization through enhanced hydrophobic interactions, achieving near-complete encapsulation efficiency (~100%) and excellent serum stability. In contrast, dendrimers with fewer cholic acids or different hydrophobic groups, such as PLCA2 or PEG-lysine-histidine variants, failed to maintain NP integrity under physiological conditions. These findings demonstrate that dendrimer architecture, particularly the degree and nature of hydrophobic conjugation, plays a decisive role in coordinating stable metal–drug complexation at high loading, a recurring challenge in nanomedicine. To support clinical translation, the manufacturability and pharmacokinetic behavior of the NP formulation were critically evaluated. Batch-to-batch reproducibility was confirmed across three independent syntheses of the polymers including PLCA8, showing consistent physicochemical characteristics with minimal variation in yield (±5%). The [Cu(DDC)_2_] NP maintained a consistent particle size and exhibited stable surface charge with low variability.

In vitro studies confirmed that [Cu(DDC)_2_] NP exhibited high potency against TNBC cells, with a notably low IC_50_ value. Treatment with [Cu(DDC)_2_] NP induced marked oxidative stress and ER stress in these cells. Mechanistically, [Cu(DDC)_2_] is known to inhibit the NPL4-p97 segregase complex, a key regulator of ERAD [21]. This inhibition impairs proteostasis by preventing the clearance of misfolded and polyubiquitinated proteins, leading to sustained ER stress and proteotoxic accumulation [22,23]. When coupled with excessive ROS generation, these stressors can push tumor cells into a non-apoptotic form of cell death known as paraptosis, a mechanism previously observed in other cancer models treated with [Cu(DDC)_2_]. Our in vitro studies in TNBC cells revealed hallmark features of this disruption, including the buildup of polyubiquitinated proteins and upregulation of HSP70 and CHOP, markers associated with the unfolded protein response. Concurrently, the NP triggered pronounced oxidative stress, as evidenced by increased ROS and lipid peroxidation (MDA production). These stress signals are consistent with a paraptosis-like form of cell death, characterized by ER dilation and vacuolization rather than caspase-dependent apoptosis [24]. This is particularly advantageous for TNBC, where apoptosis resistance limits the efficacy of many standard chemotherapies.

Beyond direct cytotoxicity, [Cu(DDC)_2_] NP effectively induced ICD, a feature rarely achieved by small-molecule therapies alone. We observed three canonical hallmarks of ICD: translocation of CRT to the tumor cell surface, ATP release, and HMGB1 secretion [25]. These DAMPs are critical for activating DCs and priming anti-tumor immunity. Indeed, bone marrow-derived DCs co-cultured with [Cu(DDC)_2_]-treated tumor cells showed robust upregulation of CD80 and CD86, confirming functional maturation [8]. This translated downstream into adaptive immune activation, evidenced by increased IL-12 secretion, elevated IFN-γ production by T cells, and significant upregulation of Granzyme B and Perforin, two cytolytic effectors essential for CTL-mediated killing [26]. The abrogation of IFN-γ by ROS scavenger NAC highlights the mechanistic link between oxidative stress and immune activation. These findings suggest that [Cu(DDC)_2_] NPs serve not only as chemotoxic agents but also as immune adjuvants capable of overcoming tumor immune evasion.

Importantly, the therapeutic efficacy extended beyond immune modulation. [Cu(DDC)_2_] NP inhibited EMT and angiogenesis, two characteristic features of metastatic progression. Treated tumors displayed increased E-cadherin expression with concomitant suppression of N-cadherin and vimentin, indicating MET reprogramming [24]. Additionally, key pro-angiogenic and pro-metastatic mediators, including VEGF, PDGF, angiopoietin-1, MMP-3, and MMP-9, were downregulated following treatment [25,26]. These effects suggest that [Cu(DDC)_2_] NP disrupted the tumor–stroma crosstalk and prevented vascular remodeling and ECM degradation necessary for metastatic dissemination. Such microenvironmental modulation likely synergizes with ICD-induced immune activation to produce a durable anti-tumor response. Consistent with our previous investigations, the dose range used in this study aligns well with the established safety limits for copper-containing therapeutics, supporting its translational feasibility.

Taken together, the [Cu(DDC)_2_] NP platform developed here exemplifies a rationally designed, multifunctional therapeutic that couples drug repurposing with advanced materials engineering. The convergence of proteotoxic stress, ICD induction, EMT suppression, and immune reprogramming within a single formulation highlights the therapeutic versatility of this system for treating aggressive, immune-cold tumors such as TNBC (Figure 6).

## 5. Conclusions

In conclusion, our study presents a dendrimer-stabilized [Cu(DDC)_2_] NP platform that achieves high drug loading, excellent serum stability, and potent anti-tumor activity through integrated chemical and immunological mechanisms. The formulation effectively harnesses the NPL4-p97 axis to induce proteotoxic stress and ICD, while also remodeling the tumor microenvironment by suppressing EMT and angiogenic signaling. By uniting direct cytotoxicity with immune activation and metastasis inhibition, [Cu(DDC)_2_] NP offers a compelling therapeutic strategy for TNBC. The favorable safety profile, ease of synthesis, and modularity of the dendrimer design also support future translation, including combination with checkpoint inhibitors or use in other malignancies marked by proteasome or ERAD dependency.

## Figures and Tables

**Figure 1 pharmaceutics-17-01448-f001:**
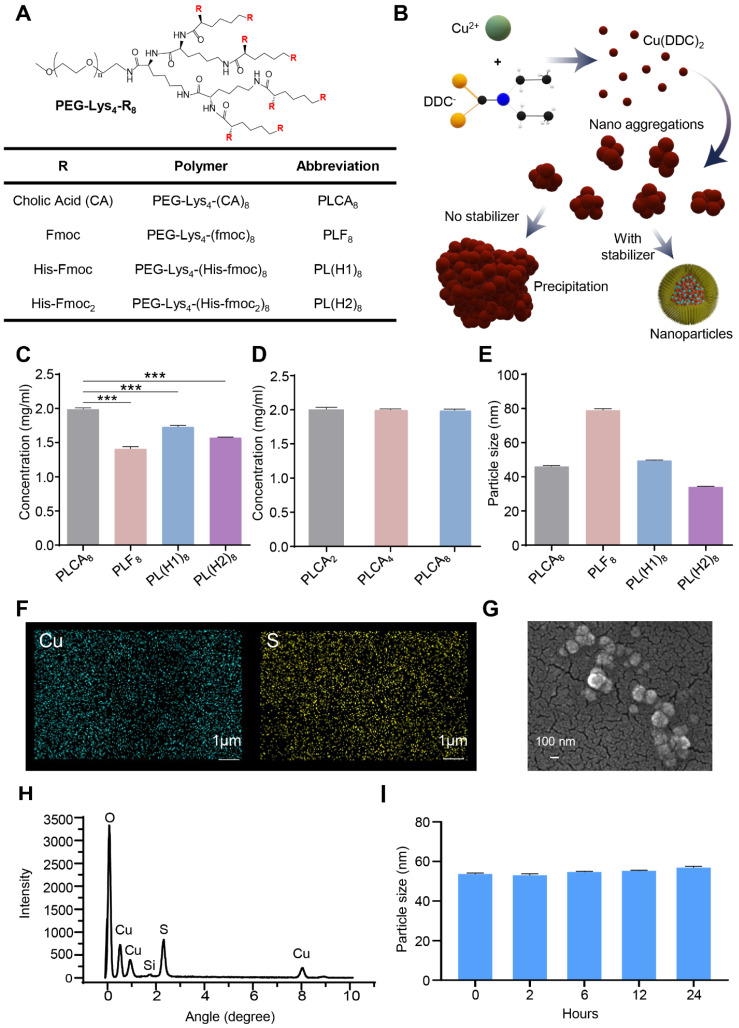
Characterization of [Cu(DDC)_2_] NP synthesized with different polymeric stabilizers. (**A**) Schematic illustration of the PEG-lysine dendrimer structure. (**B**) Diagram showing the critical role of stabilizers in the synthesis of [Cu(DDC)_2_] NP using the SMILE method. (**C**) Quantification of [Cu(DDC)_2_] loading within NP formulated using different stabilizers. Data are presented as mean ± SD (*n* = 3). (***, *p* < 0.001) (**D**) Concentrations of [Cu(DDC)_2_] encapsulated in NP prepared with different cholic acid-conjugated polymers. Data are presented as mean ± SD (*n* = 3). (**E**) The particle size of [Cu(DDC)_2_] NP prepared with various stabilizers. Data are presented as mean ± SD (*n* = 3). (**F**) EDX confirming the elemental composition of [Cu(DDC)_2_] NP. (**G**) SEM image showing the surface morphology of [Cu(DDC)_2_] NP. (**H**) EDX spectrum of [Cu(DDC)_2_] NP. (**I**) Size change of [Cu(DDC)_2_] NP prepared with PLCA8 polymer. Data are presented as mean ± SD (*n* = 3).

**Figure 2 pharmaceutics-17-01448-f002:**
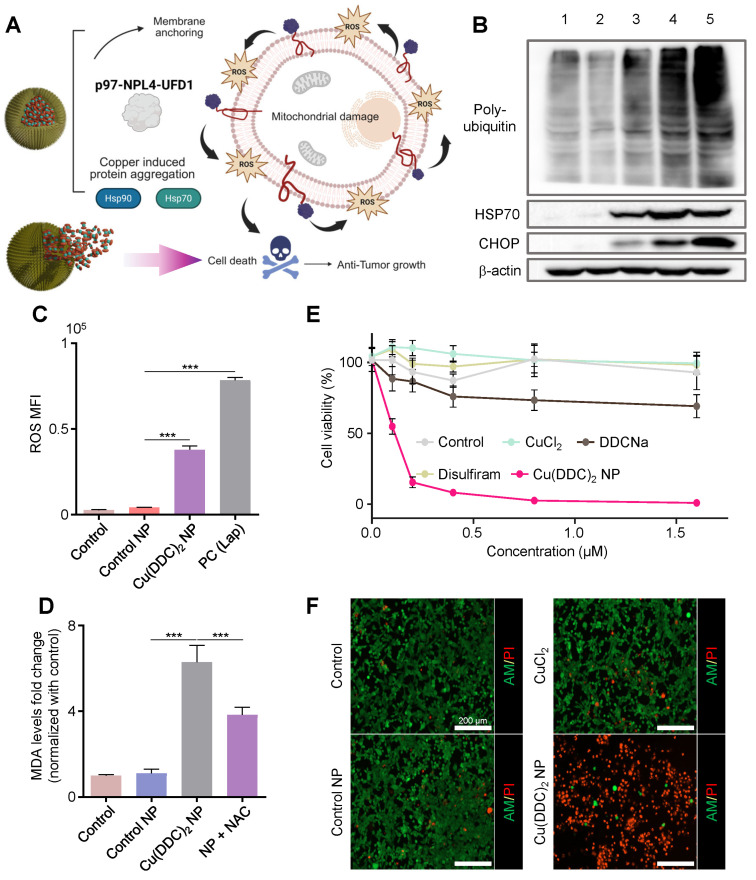
[Cu(DDC)_2_] NP anti-tumor activities and mechanism. (**A**) Schematic of the mechanisms of anticancer activity by [Cu(DDC)_2_] NPs. Created in BioRender. Kang, X. (2025) https://BioRender.com/xknyt0m (accessed on 30 October 2025) (**B**) Western blot analysis showing poly-ubiquitylated proteins, HSP 70, and CHOP levels in different treatment groups. Lane 1: Blank control (untreated). Lane 2: Control NP (PLCA8 only). Lane 3: [Cu(DDC)_2_] 1 µM in dimethyl sulfoxide. Lane 4: [Cu(DDC)_2_] NP 1µM. Lane 5: Bortezomib (0.1 µM, positive control). β-actin was used as the loading control. (**C**) The ROS levels of the cells after treating with PLCA8 (Control NP), [Cu(DDC)_2_] NP, and Lapatinib (Positive control). Data are presented as mean ± SD (*n* = 3). (***, *p* < 0.001) (**D**) Measurement of lipid peroxidation using the MDA assay. [Cu(DDC)_2_] NP treatment markedly elevated MDA levels, indicating oxidative damage to cellular membranes. Co-treatment with the antioxidant NAC significantly attenuated MDA accumulation. Data are presented as mean ± SD (*n* = 3). (***, *p* < 0.001) (**E**) 4T1 cells were treated with different formulations for 48 h and analyzed with the MTT assay. Data are presented as mean ± SD (*n* = 3). (**F**) 4T1 cells were treated with PBS (control), PLCA8 1 µM (control NP), CuCl_2_ 1 µM, and [Cu(DDC)_2_] NP 1 µM for 24 h. Then cells were stained with Calcein-AM (green) and PI (red).

**Figure 3 pharmaceutics-17-01448-f003:**
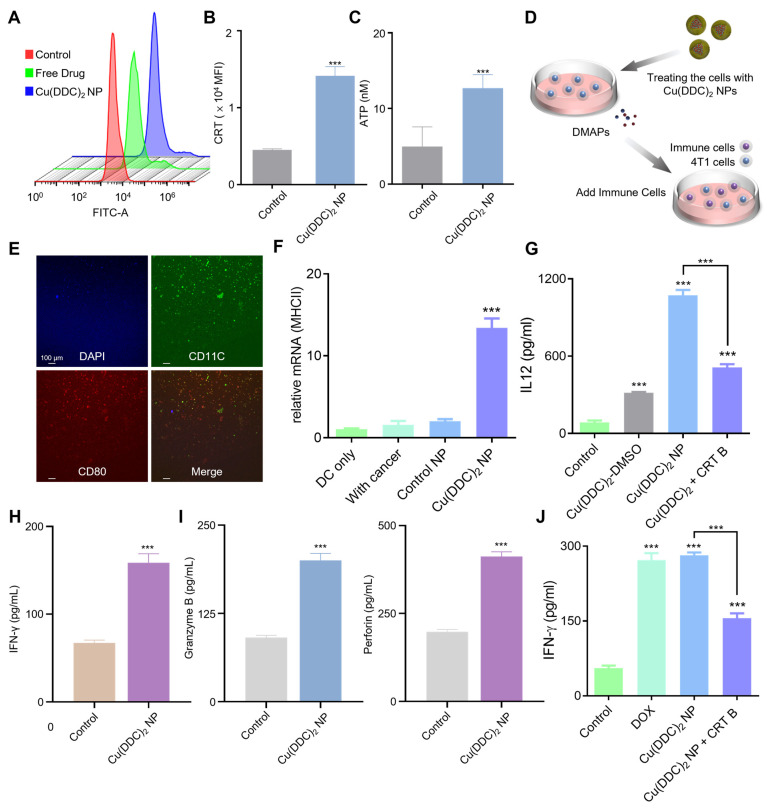
[Cu(DDC)_2_] NP induced ICD and promoted anti-tumor immune responses. (**A**) Flow cytometry analysis of CRT exposure on the cell surface after treating with Control, [Cu(DDC)_2_] in DMSO (free drug), and [Cu(DDC)_2_] NP. (**B**) Quantification of surface CRT expression presented as mean fluorescence intensity (MFI). (**C**) ATP release in the culture supernatant was measured by the ATP determination kit. (**D**) Schematic illustration of the DCs maturation assay. Tumor cells were treated with [Cu(DDC)_2_] NP to generate DAMPs, followed by co-culture with immune cells. (**E**) Immunofluorescence staining of bone marrow-derived dendritic cells (BMDCs) showing co-expression of activation marker CD80 (red) and dendritic cell marker CD11c (green), with nuclei stained by DAPI (blue). (**F**) The MHCII mRNA level was determined by qPCR. (**G**) Enzyme-linked immunosorbent assay (ELISA) quantification of IL12, (**H**) Enzyme-linked immunosorbent assay (ELISA) quantification of IFN-γ, (**I**) Levels of Granzyme B, and Perforin levels after treating with [Cu(DDC)_2_] NP. (**J**) Levels of IFN-γ following different treatments. Data are presented as mean ± SD (*n* = 3). (***, *p* < 0.001).

**Figure 4 pharmaceutics-17-01448-f004:**
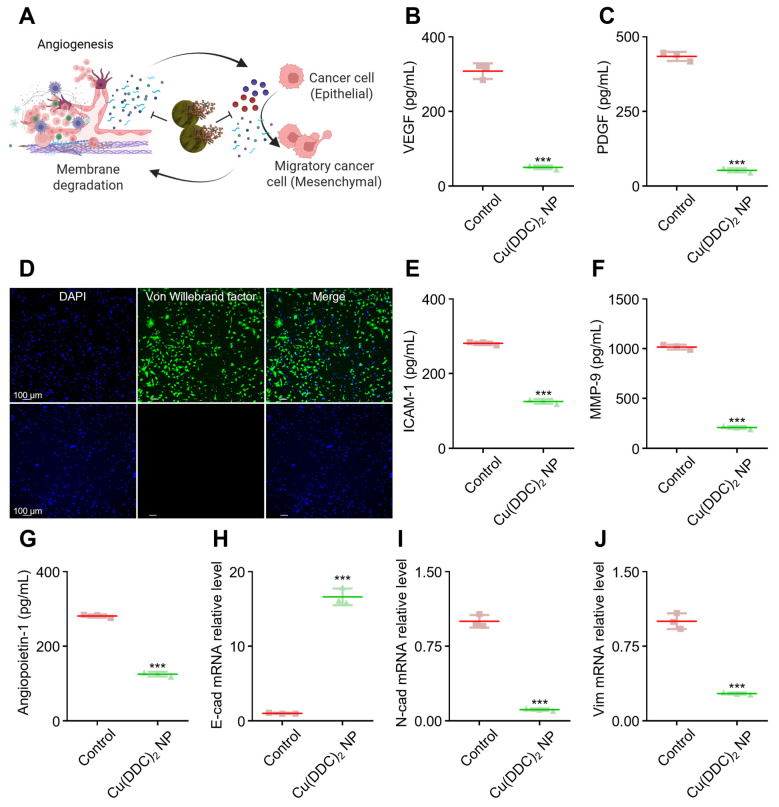
[Cu(DDC)_2_] NP inhibited angiogenesis and suppressed epithelial–mesenchymal transition (EMT) in the tumor microenvironment. (**A**) Schematic representation showing how [Cu(DDC)_2_] NP disrupts angiogenic activation and EMT, both of which are critical for cancer metastasis. Created in BioRender. Kang, X. (2025) https://BioRender.com/61j3qzn (accessed on 30 October 2025) (**B**) ELISA quantification of VEGF levels and (**C**) PDGF levels in the co-culture of cancer cells and endothelial cells (ECs) following [Cu(DDC)_2_] treatment. (**D**) Immunofluorescence staining of von Willebrand factor (green) and DAPI (blue) in ECs. Top: Control; Bottom: [Cu(DDC)_2_] NPs-treated. (**E**) The ICAM-1, (**F**) MMP-9, (**G**) Angiopoietin-1 levels in co-culture cells were measured by ELISA with or without [Cu(DDC)_2_] NP treatment. (**H**) Relative mRNA expression of E-cadherin (epithelial marker), (I) N-cadherin (mesenchymal marker), and (**J**) Vimentin in cancer cells after treatment with [Cu(DDC)_2_] NP. Data are presented as mean ± SD (*n* = 3); *** *p* < 0.001.

**Figure 5 pharmaceutics-17-01448-f005:**
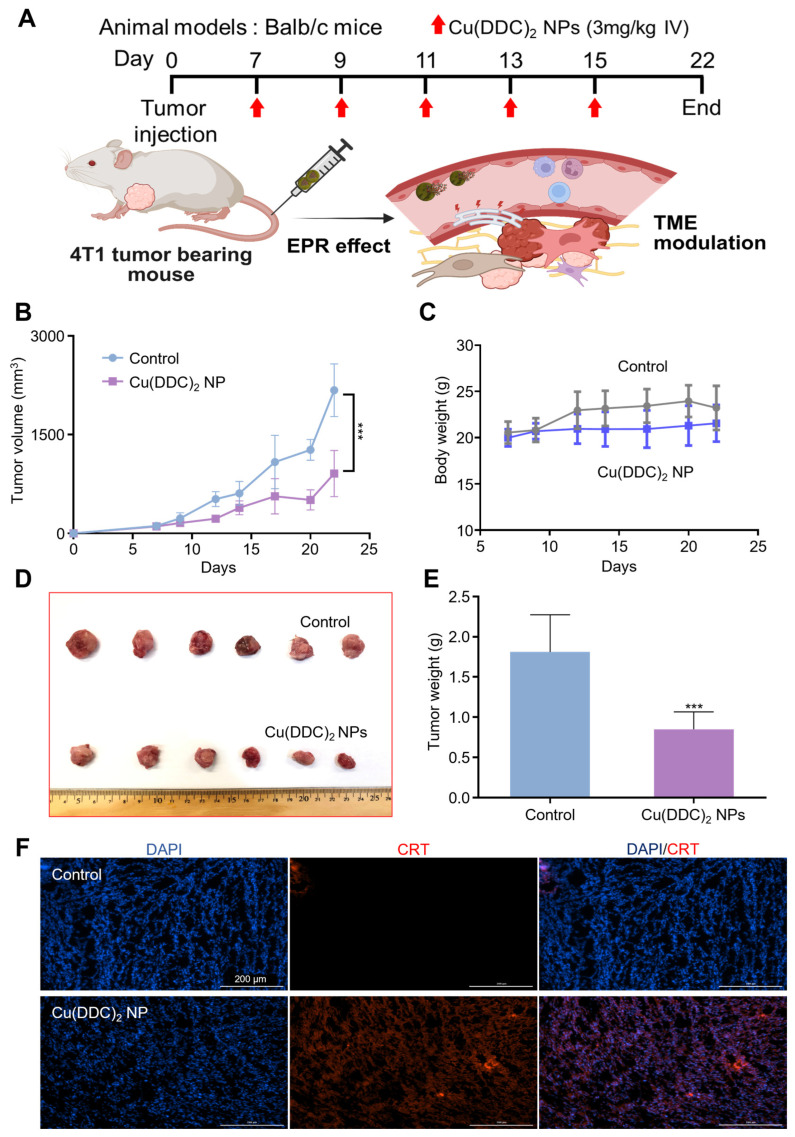
The in vivo anticancer activity of [Cu(DDC)_2_] NP. (**A**) Experimental design of in vivo anti-tumor activities of [Cu(DDC)_2_] NP. Created in BioRender. Kang, X. (2025) https://BioRender.com/xknyt0m (accessed on 30 October 2025). (**B**) Tumor growth of 4T1 cells in BALB/c mice injected with 10% PLCA8 in PBS (control) or [Cu(DDC)_2_] NP as indicated in (**A**). Data are presented as mean ± SD (*n* = 6). (***, *p* < 0.001) (**C**) The change of body weight of tumor bearing mice models during the treatment. Data are presented as mean ± SD (*n* = 6). (***, *p* < 0.001) (**D**) Images of collected tumors (**E**) The measurement of tumor weight at the end of the study. Data are presented as mean ± SD (*n* = 6). (***, *p* < 0.001). (**F**) Levels of CRT expression on tumor tissue which collected from 4T1 BALB/c mice.

**Figure 6 pharmaceutics-17-01448-f006:**
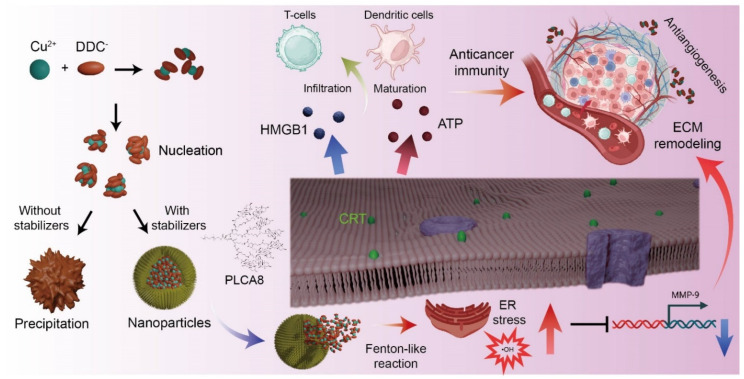
Schematic of [Cu(DDC)_2_] NP-Induced Immunogenic Cell Death and Tumor Microenvironment Modulation. Stabilized [Cu(DDC)_2_] NP induces ER stress and Fenton-like reactions, leading to ICD via DAMPs release. These effects promote DC maturation, T cell infiltration, and remodel the tumor microenvironment via anti-angiogenesis and MMP-9 suppression. Part of figures were created in BioRender. Kang, X. (2025) https://BioRender.com/61j3qzn (accessed on 30 October 2025).

## Data Availability

The original contributions presented in this study are included in the article/Appendix A. Further inquiries can be directed to the corresponding authors.

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
