# Peer review of "Novel Disulfiram-Loaded Metal–Organic Nanoparticles Inhibit Tumor Growth and Induce Immunogenic Cell Death of Triple-Negative Breast Cancer Cells"

_pharmaceutics, 2025, doi:10.3390/pharmaceutics17111448_

Round 1
Reviewer 1 Report
Comments and Suggestions for Authors
-
The description of the PLCA8 dendrimer (synthesis and structure) lacks quantitative characterization. please report the molecular weight distribution (SEC or MALDI-TOF), the average degree of cholic-acid substitution per PEG-lysine scaffold, and NMR confirmation (1H/13C) of substitution. these data are essential to reproduce the formulation and to interpret loading/stability differences between PLCA variants (methods section, synthesis steps).
-
There are contradictory or unclear statements about particle size and stability (text reports 40–80 nm, then says PLCA8 displayed larger sizes ~80 nm and was excluded). provide complete DLS data including intensity/number distributions, PDI, raw size histograms, and the number of technical/biological replicates. include TEM or cryo-TEM micrographs (not only SEM) to show internal morphology and to confirm single-particle measurements in hydrated state. report stability time-courses numerically (size and concentration vs time) with statistical measures.
-
The claim of “~100% loading efficiency” needs stronger analytical validation. the UV assay at 435 nm may be confounded by matrix/stabilizer absorption—please quantify payload and copper by orthogonal methods (HPLC with validated extraction, and ICP-MS or AAS for Cu content). report both percent encapsulation efficiency and absolute drug loading (mg drug per mg NP), limit of detection, calibration curves, and how free vs bound drug was separated (centrifugation/filtration method validation).
-
In vitro exposure and control conditions require clarification and standardization. clearly state how free Cu(DDC)2 (poorly soluble) was prepared and solubilized for the “free drug” controls, and confirm that dosing is molar-equivalent between NP and free drug groups. reconcile differing assay time points (MTT 48 h vs CRT 12 h vs ATP 12 h vs Calcein/PI 24 h) with a short rationale for each endpoint and show time-course data for key readouts. explicitly state whether n=3 refers to biological or technical replicates and include exact p-values and multiple comparisons corrections where appropriate.
-
Mechanistic claims invoking NPL4–p97 sequestration and proteasome dysfunction are plausible but under-supported. include direct evidence of NPL4 aggregation (immunofluorescence co-localization, detergent-insoluble fractionation, or co-IP), proteasome activity assays (e.g., fluorogenic peptide substrates) and time-resolved ubiquitin accumulation. rescue experiments (e.g., NAC, proteasome inhibitors, or NPL4 overexpression/knockdown) would strengthen causal statements. provide representative uncropped western blots with molecular weight markers in supplementary.
-
The evidence for ICD is promising but incomplete for a translational claim. provide full flow cytometry histograms and gating strategy for CRT; show representative raw data for HMGB1 ELISA including standard curve and normalization to cell number; and include an established positive control ICD inducer (e.g., doxorubicin or oxaliplatin) run in parallel. in vivo immune readouts should go beyond CRT IF—add flow cytometry of tumor-infiltrating lymphocytes (CD8+, CD4+, Tregs, myeloid subsets), and cytokine profiling of tumor homogenates or serum to show functional immune engagement.
-
In vivo experimental design and reporting need tightening. clarify the dosing schedule (manuscript contains more than one regimen in different sections), define whether 3 mg/kg refers to Cu, DDC, or total complex, and provide rationale and power calculation for n=6/group. include basic safety/toxicity endpoints: complete blood chemistry, liver/kidney histopathology, and biodistribution (ICP-MS measurement of Cu in major organs) to address systemic copper exposure and off-target accumulation. report survival or longer-term follow-up if available.
-
Claims on anti-angiogenesis and EMT modulation should be supported by functional assays and orthogonal protein data. include endothelial tube-formation or migration assays, tumor cell transwell migration/invasion assays, and provide western blots for E-cadherin, N-cadherin and vimentin with quantification (densitometry normalized to loading control). ensure the ELISA kits used are validated for the species (mouse) and provide standard curves and technical reproducibility data.
-
Consider adding a dedicated translational/discussion subsection addressing formulation manufacturability and pharmacokinetics: comment on batch-to-batch reproducibility, potential copper release kinetics from NP (in plasma and simulated fluids), predicted human equivalent doses, and clearance pathways. if possible, include preliminary PK/biodistribution data (plasma Cu levels over time, tumor vs liver/kidney accumulation) or at minimum discuss limitations and safety risks associated with systemic copper delivery.
Consistent terminology and abbreviation use: define each abbreviation at first use and then use it consistently (avoid switching between full name and acronym). Also harmonize naming (e.g., NP vs nanoparticle, PLCA8) and the formatting of chemical/drug names throughout the text.
Improve sentence clarity and length: several paragraphs contain very long, complex sentences that obscure the meaning — break these into shorter sentences (one idea per sentence) and move subordinate details into separate clauses or sentences for easier reading.
Reviewer 2 Report
Comments and Suggestions for Authors
The manuscript “Novel Disulfiram-Loaded Metal-Organic Nanoparticles Inhibit Tumor Growth and Induce Immunogenic Cell Death of Triple Negative Breast Cancer Cells” present a clear and scientifically sound study on design, preparation, characterisation and cytotoxicity against Triple Negative Breast Cancer Cells of a [Cu(DDC)2] NPs obtained by applying the SMILE approach and using an amphiphilic PLCA bearing cholic acid moieties.
The main aspect addressed by the research consist in finding a method in order to develop a proper biocompatible delivery system that replace an old antitumor drug disulfiram through the Cu(II) complex with its metabolite, species embedded in an organic matrix.
The topic is relevant for the field having in view that bring a new formulation for an old drug of which administration is accompanied by inconveniences like in situ [Cu(DDC)2] generation, the short plasma half-life and rapid systemic metabolization. This formulation allows a new way of administration that improve the antitumor activity and solve some aspects concerning the [Cu(DDC)2] both solubility and cell accumulation. An important aspect comes from the fact that formulation exhibit activity in sub-micromolar level, and this activity is preserved in vivo as well. Moreover, in vivo assay is important for develop an efficient antitumor species more over that this is not addressed in many researches in the field, majority of them being limited to in vitro ones.
The authors obtained a NP formulation with high encapsulation efficiency, enhanced serum stability, and potent cytotoxicity and moreover succeeded to improve both the solubility and accumulation into tumor cells. Moreover, this species proved in vivo efficacy.
The references are appropriate and helpful both for the paper topic and data discussed in paper.
The paper is well-organized and well written and conclusions are sound and justified by the data presented.
The figures display a high resolution and are suggestive both for the experimental protocol and for proposed mechanism of action.
As result, my overall comment is that these data present interest concerning the importance of finding new species for the treatment of this aggressive subtype of cancer with earlier metastasis, overall poorer prognosis, and that rapidly develop resistance.
Maybe the NP cytotoxicity on healthy cells could improve the paper and bring an inside into the selectivity index of this formulation, which should have a reduced toxicity in comparison with [Cu(DDC)2] alone.
I therefore recommend publishing after minor corrections such as:
- The compound must be written as complex, namely [Cu(DDC)2] in whole paper.
- The in vivo and in vitro must be provided in Italic style in whole paper.
- The ml must be corrected as mL.
Author Response
Thank you , please see attached

Reviewer 3 Report
Comments and Suggestions for Authors
Major Revision
This manuscript describes the development of dendrimer-stabilized Cu(DDC)₂ nanoparticles for treating triple-negative breast cancer (TNBC). The authors provide a comprehensive characterization of the nanoparticle system and present mechanistic in-vitro and in-vivo evidence for its cytotoxic, immunogenic, and anti-metastatic effects.
Please address following concerns:
1] Typographical errors animo @page-3 line-112, mou44se @page-5 line-221.
2] The introduction is somewhat lengthy and could be tightened to focus more quickly on the nanoparticle rationale.
3] Lot of mechanism supportive data is correlative: proteasome disruption and ROS elevation are suggestive of ER stress, but direct visualization of ER dilation/paraptosis by TEM is missing. Similarly, ICD is inferred from CRT, ATP, and HMGB1 markers, but functional assays are lacking. Do caspase assays to rule out apoptosis.
4] Immune activation in-vivo is only partially supported (CRT staining in tumors). Flow cytometry of tumor-infiltrating lymphocytes would be more convincing.
5] The efficacy of Cu(DDC)₂ NP is compared mainly to free drug or blank NP. However, clinically relevant benchmarks (e.g. doxorubicin) are missing.
6] Contradictory descriptions of PLCA8 particle sizes: text states PLCA8 gave ~80 nm @ page-7 line-278, then PLCA8 shows ~40 nm page-7 line-283 and was used.
7] No MALDI-TOF done to confirm molecular weight, degree of cholic-acid substitution, PEGylation, purity, and polydispersity of the dendrimer (PLCA8). Particle characterization relies on DLS and FESEM only; no TEM (needed for hydrated morphology), no zeta potential.
8] Encapsulation/loading claims are reported based on UV 435 nm on a Nanodrop. This is not a robust measurement for metal complexes and is subject to interference from polymer absorbance and scattering.
9] It lacks PK/biodistribution by ICP-MS (plasma/major organs/tumor) and serum protein binding analysis.
Round 2
Reviewer 3 Report
Comments and Suggestions for Authors
The authors have addressed all comments thoroughly. Please accept the manuscript for publication.